# Implementation of a goal-directed Care Bundle for intracerebral hemorrhage: Results of embedded process evaluation in the INTERACT3 trial

Menglu Ouyang[1,2]*, Anila Anjum[3], Francisca Gonzalez Mc Cawley[4], Mohammad Wasay[3], Lu Ma[5], Xin Hu[5], Xiaoying Chen[1], Alejandra Malavera[1], Xi Li[5], Paula Muñoz Venturelli[1,4], H. Asita de Silva[6], Nguyen Huy Thang[7], Kolawole W. Wahab[8], Jeyaraj D. Pandian[9], Octavio M. Pontes-Neto[10], Carlos Abanto[11], Venessa Cano-Nigenda[12], Antonio Arauz[12], Chao You[5], Stephen Jan[1], Lili Song[13]*, Craig S. Anderson[1,13,14,15], Hueiming Liu[1,16,17], for the INTERACT3 Investigators

1 The George Institute for Global Health, Faculty of Medicine, University of New South Wales, Sydney, Australia, 2 The George Institute China Office, Beijing, China, 3 Clinical Trials Unit, The Aga Khan University, Karachi, Pakistan, 4 Centro de Estudios Clínicos, Instituto de Ciencias e Innovación en Medicina, Facultad de Medicina Clínica Alemana Universidad del Desarrollo, Chile, 5 Department of Neurosurgery, West China Hospital, Sichuan University, Chengdu, China, 6 Clinical Trials Unit, Faculty of Medicine, University of Kelaniya, Kelaniya, Sri Lanka, 7 Stroke Unit, 115 Hospital, Ho Chi Minh City, Vietnam, 8 Department of Medicine, University of Ilorin & University of Ilorin Teaching Hospital, Ilorin, Nigeria, 9 Department of Neurology, Christian Medical College and Hospital, Ludhiana, India, 10 Department of Neurology, Ribeirão Preto Medical School, University of São Paulo, Ribeirao Preto, Brazil, 11 Cerebrovascular Disease Research Center, National Institute of Neurological Sciences, Lima, Peru, 12 Stroke Clinic, Instituto Nacional de Neurología y Neurocirugía Manuel Velasco Suarez, Mexico City, Mexico, 13 Institute of Science and Technology for Brain-Inspired Intelligence, Fudan University, Shanghai, China, 14 Neurology Department, Royal Prince Alfred Hospital, Sydney Health Partners, Sydney, Australia, 15 Heart Health Research Center, Beijing, China, 16 Menzies Centre for Health Policy and Economics, University of Sydney, Sydney, Australia, 17 Sydney Institute for Women, Children and their Families, Sydney Local Health District, Sydney, Australia

* mouyang@georgeinstitute.org.au (MO); lili_song@fudan.edu.cn (LS)

Data Availability Statement: Individual, de-identified participant data used in these analyses will be shared on request from any qualified

## Abstract

The third, stepped-wedge, cluster-randomized, Intensive Care Bundle with Blood Pressure Reduction in Acute Cerebral Hemorrhage Trial (INTERACT3), has shown that a goal-directed multi-faceted Care Bundle incorporating protocols for the management of physio-logical variables was safe and effective for improving functional recovery in a broad range of patients with acute intracerebral hemorrhage (ICH). The INTERACT3 Care Bundle included time- and target-based protocols for the management of early intensive lowering of systolic blood pressure (SBP, target <140mmHg), glucose control (target 6.1–7.8 mmol/L in those without diabetes and 7.8–10.0 mmol/L in those with diabetes), anti-pyrexia treatment (target body temperature $\leq$37.5˚C), and the rapid reversal of warfarin-related anticoagulation (target international normalized ratio <1.5). An embedded process evaluation was conducted to allow a better understanding of how the Care Bundle was implemented in different countries to enhance the transferability of this evidence in the international context. This study used a mixed-methods approach involving interviews, focus group discussions, and surveys to evaluate the implementation outcomes included fidelity, dose, reach, acceptability, appropriateness, adoption, and sustainability. Interviews (n = 27), focus group discussions (n = 3),

investigator after the approval of a protocol and signed data access agreement via both the trial steering committee and the research office of The George Institute for Global Health (Sydney, NSW, Australia). All the data sharing to external need to go through a formal request to Data Sharing Committee (DSC) DSC@georgeinstitute.org.au to access data. Data sharing policy guidance: https://www.georgeinstitute.org/data-sharing-policy.

**Funding:** The study is supported by multiple funders, including joint fund provided by the Department of Health and Social Care, the Foreign, Commonwealth & Development Office, the Medical Research Council and the Welcome Trust (all London, UK, grant reference number MR/T005009/1), the West China Hospital Outstanding Discipline Development 1–3-5 programme grant (number ZY2016102), National Health and Medical Research Council of Australia grant (number APP1149987) and Sichuan Credit Pharmaceutical and Takeda (China). The funders had no role in study design, data collection and analysis, decision to publish, or preparation of the manuscript.

**Competing interests:** I have read the journal's policy and the authors of this manuscript have the following competing interests: LS reports funding from the Medical Research Council of the UK, Sichuan Credit Pharmaceutic, and Takeda China; and speaker fees from Takeda China. CSA has received grants from the National Health and Medical Research Council and Medical Research Futures Fund of Australia, the Medical Research Council of the UK, Penumbra, and Takeda China. CY has received funding from West China Hospital. The other authors have declared that no competing interests exist.

and quantitative surveys (n = 48) were conducted in 7 low- and middle-income countries (LMICs) and 1 high-income country during 2019–2022. The Care Bundle was generally delivered as planned and well accepted by stakeholders, although some difficulties were reported in reaching the SBP and glycemic targets. Contextual factors including staff shortage, limited availability of antihypertensive drugs, and delayed systems of care processes, were common barriers to implementing the Care Bundle. Facilitating factors included good communication and collaboration with staff in emergency departments, the development of pathways within available resources, and regular training and monitoring. Our process evaluation provides useful insights into the contextual barriers which need to be addressed for effective scale up of the Care Bundle implementation in a global context.

**Trial registration**: INTERACT3 is registered at Clinicaltrials.gov (NCT03209258) and the Chinese Clinical Trial Registry (ChiCTR-IOC-17011787).

## Introduction

Intracerebral hemorrhage (ICH) is the most serious and least treatable form of stroke, accounting for approximately 20% of the 20 million new strokes that occur in the world each year [1]. However, the burden of ICH is the greatest in low- and middle-income countries (LMIC) where the incidence is very high due to the high prevalence of hypertension, unhealthy diets and other risk factors [1], and there is limited access to surgical or medical treatments [2]. Hematoma expansion (HE) is a strong predictor of disability and mortality after ICH, with every increase of 1 ml in absolute haematoma volume, it leads to a 7% greater likelihood for patients to shift from independence to disability [3]. HE occurs mostly within 3 hrs from the time of ICH onset and stops progressing after 24 hrs [4]. Therefore, timely control of HE is an important therapeutic target in ICH management. For many years, clinical trials that focus on a single intervention to control HE have failed to establish a specific beneficial treatment after ICH. This has led to therapeutic nihilism in the approach to ICH care [5].

Since blood pressure, glycaemia and temperature are often altered from impaired autoregulation after ICH and are associated with haematoma expansion, death and disability, the active management for ICH should target on early management of these physiological parameters. A bundled care approach with a set of evidence-informed interventions that are promising to improve patient outcomes. The Australian Quality in Acute Stroke Care (QASC) trial was the first to show that a nursing protocol for the management of fever, hyperglycaemia, and poor swallow improved outcomes for patients with mixed types of stroke [6]. The single-centre before-after Acute Bundle of Care for Intracerebral Haemorrhage (ABC-ICH) found that combined interventions including anticoagulation reversal and BP control, and immediate access to neurosurgical and critical care services reduced 10.8% of the case fatality at 30 days [7]. However, the previous studies were conducted in Australia and UK, with no evidence of the effectiveness and implementation of the bundled care approach management for ICH in LMICs which has a higher incidence of ICH and different health system contexts. Considering the limited resources such as a lack of neurosurgical settings and speech pathology expertise for dysphagia/swallowing screening in the LMICs, we focused on the management of physiological parameters in our third Intensive Care Bundle with Blood Pressure Reduction in Acute Cerebral Hemorrhage Trial (INTERACT3). Our study has extended the knowledge of the benefits of bundled care, which was specifically undertaken to determine if a simple, multi-

faceted, Care Bundle incorporating time- and target-based protocols for the management of abnormal physiological variables could improve outcomes for patients with ICH in a broad range of health care settings [8, 9]. Using an international, multicenter, stepped-wedge, cluster-randomized controlled design, the Care Bundle has shown to improve functional recovery (common odds ratio 0.86, 95% confidence interval 0.76 to 0.97; P = 0.02) in 7036 patients enrolled at 121 hospitals in nine LMIC (Brazil, China, India, Mexico, Nigeria, Pakistan, Peru, Sri Lanka, and Vietnam), and one high-income country (Chile) between December 12, 2017, and December 31, 2021.

To successfully implement and adapt the Care Bundle in the real world to benefit a broader population [10], it is important to understand how the implementation of this complex intervention with multiple components was undertaken with the necessary behavioural and organisational changes in local structures and processes of care [11, 12]. A process evaluation (PE) was embedded within INTERACT3 to examine implementation outcomes and provide explanations for discrepancies between expected and observed outcomes. The approach offers insights into how contextual factors influence outcomes for considering the ongoing and wider implementation of the Care Bundle [13]. We have already published results of the PE conducted in China where most of the participating sites and patients were located; we noted implementation difficulties that perceived concerns about how the protocol-defined BP and glycemic targets might harm patients, and health system contextual issues related to staffing and medication supply [14]. Understanding facilitators, barriers, and mechanisms in integrating an intervention in complex system, is essential for quality improvement and scale up [15], we herein report results of implementation outcomes for countries outside of China and the various challenges to integrating the Care Bundle into routine care, and explore factors for generalizability and sustainability for the wider implementation of this system of care.

## Methods

### Design, setting, and participants

Our PE was informed by the Medical Research Council (MRC) [11] recommendations and an implementation research logic model [16] to include the following components: implementation outcomes (reach, acceptability and appropriateness, fidelity, dose, adoption, and sustainability), mechanisms of impact, and contextual factors. Normalisation process theory [17] was used to understand the mechanisms of the integration of the Care Bundle into routine practice across the core domains of coherence, cognitive participation, collective action, and reflexive monitoring. Details of the design are published elsewhere [13]. In brief, the study was undertaken in the dedicated areas for the care of ICH patients in participating hospitals such as the emergency department, acute stroke unit, intensive care unit, and neurology and neurosurgery areas/wards. Key stakeholders included healthcare providers who were involved in delivering the intervention and patients who received the Care Bundle and their family members/carers. Since the trial was initiated earlier in China in December 2017 than in other countries, and the process evaluation for the sites in China was submitted before the study conduction in other countries, this study will mainly focus on the process evaluation in other countries and compare their findings to those findings from China.

### Data collection

A mixed-methods approach included quantitative (questionnaire, survey, case report forms and monitoring logs) and qualitative (focus group discussion and semi-structured interviews) data sources used to evaluate the implementation of interventions and interpretation of the meaningfulness of outcomes in the trial [18]. A feasibility hospital organization questionnaire

distributed to all hospitals for trial start-up activities was used to define the characteristics of sampled hospitals for capturing health system contexts. The rate of patient recruitment was obtained from monitoring logs and data entry reports to inform site sampling selection for the interviews. A quality control survey was collected to understand the experience and project operational barriers to implementing the intervention at sites in Brazil, Chile, India, Mexico, Nigeria, Pakistan, Peru, Sri Lanka, and Vietnam in September 2021- February 2022. Focus group discussions were conducted with site principal investigators (PIs) and national leaders in South America (2 PIs), Nigeria (4 PIs), Pakistan (2 PIs) and Vietnam (3 PIs) through the teleconference/video conference, prior to in-depth interviews to discuss roles and responsibilities, research experience, and how the intervention was implemented at an organizational level. Each of the focus group discussions was 30–40 minutes with audio recorded and facilitated by an experienced researcher at the international coordinating centre. Semi-structured interviews were conducted by standard trained researchers from Regional Coordinating Centers, with audio recordings. During the interview process, questions were iteratively modified to allow a deeper exploration of emergent themes identified by the research staff. All interviews were conducted in local languages either online via teleconference or in-person, according to hospital policies in relation to the COVID-19 pandemic and availability.

## Sampling

Purposive sampling was used to select representative sites for semi-structured interviews [19]: this was based on pre-specified criteria (geographical location, hospital level, department, rate of patient recruitment, and data entry quality) and stratified by country, to achieve representativeness across participating sites. At least three clinicians (doctors/nurses/other ward staff) and two patients at each sampled site were invited to interview. Sixteen sites from nine countries were invited to participate during the early intervention phase, and 11 sites agreed to participate from India, Sri Lanka, Vietnam, Pakistan, Chile, Nigeria and Mexico.

## Data analysis and reporting

Quantitative data was analyzed using descriptive statistics and reported in numbers and percentages. Audio recordings were transcribed to de-identifiable qualitative data and analyzed independently by trained coders (MO, AA, FG) to ensure the credibility of the findings between two coders. Non-English transcripts were translated and coded in English. The initial thematic codebook was built upon the PE evaluation in China [14]. Thematic saturation was reached with no new codes being created until the first 5 interview transcripts were coded, with agreement being reached between two coders. Preliminary findings were discussed with research staff over interpretation and a need to explore the significance of the results. To better summarize the contextual factors that identified influence the care bundle implementation, we used Consolidated Framtework for Implementation Research (CFIR) to report [20]. Nvivo 11.0 was used to manage the data and the standards for reporting qualitative research (S1 Checklist) checklist was used to report the findings from focus group discussions and interviews [21].

## Results

### Participants characteristics

A total of 27 interviews (16 doctors, 3 nurses, and 8 patients/carers from 11 hospitals), 3 focus group discussions with national leaders and principal investigators, and 47 responses were received from the quality control survey to staff at 25 (64.1%) of 39 sampled hospitals, were

conducted in 2021–2022. The rates of participating sites by each survey has shown in Supplemental S1 Table. Detailed characteristics of participants in interviews and surveys are summarized in S2 Table and S1 Fig.

## Implementation outcomes

Table 1 shows the evaluation results of implementation outcomes. Findings were synthesized in Fig 1 using the Implementation Research Logic Model.

## Acceptability & appropriateness

Overall, the Care Bundle was well accepted by clinicians although they expressed concerns that more intensive control may cause harm, and reported difficulties in the early phase of

**Table 1. Implementation outcomes of the Care Bundle.**

| Implementation outcomes | Coded theme | Quote |
|---|---|---|
| Reach | • Eligible patients were recruited after admission<br>• Difficulties in recruitment due to contextual factors, such as the COVID-19 pandemic, economic burden, shortage of staff and bed resources, delayed presentation, early transfer to other hospitals and consent issues | *"Recruiting patients was difficult because most of the patients go to local doctors and there's a lot of delay in the tertiary care hospital" (India, Neurologist)*<br>*"Since clinical trial/clinical study is not familiar with Vietnamese patients/caregivers, to obtain their consents is not easy…they don't want to participate any kind of uncertain treatment or abnormal procedure" (Vietnam, Neurologist)"*<br>*"Due to the COVID pandemic, our ward staff and space facility is limited, such as the availability of high dependency units and monitoring." (Sri Lanka, Neurologist)* |
| Acceptability and appropriateness | • Clinicians had a good understanding of the rationale and perceived the benefit of the care bundle to ICH management<br>• Both clinicians and patients accepted the care bundle | *"Although it is known that BP control is beneficial in ICH, when provided as part of an acute care bundle along with control of other vital parameters like sugar, INR and temperature, the benefit could be more. I had no concerns as the aspects are known to be beneficial, not harmful." (Sri Lanka, Neurologist)*<br>*"This study tries to delve a little deeper not only into the blood pressure targets, but also into other variables that one can manage in hyperacute and in the intensive. And that seems super good to me to study it super perfect" (Chile, focus group)*<br>*"It is very wonderful all through. Honestly, I have never experienced such closeness and rapport with medical personalities and people like that so. It has been very wonderful." (Nigeria, Patient)*<br>*"Monitoring frequency was challenged to follow since it has big difference compared to routine practice and nurses were hard to follow." (Vietnam, Neurologist)* |
| Fidelity | • The Care Bundle was delivered according to the protocol<br>• Difficulty in transferring over to the intervention at the early phase | *"All the components of care bundle (have) given to the patients." (Sri Lanka, Neurologist)*<br>*"Cross-over was tough, the other healthcare staff weren't very happy to take additional patients who required intense monitoring." (Sri Lanka, Neurologist)* |
| Dose | • Monitoring of all the components of the Care Bundle, particularly of intensive BP monitoring<br>• BGL not frequently monitored<br>• Difficulty with timely achievement of the management targets<br>• Awaiting PCR results delay in initiating care bundle | *"Obvious all parameters are monitored for all the patients in this trial…BP and sugar are monitored for all patients." (Pakistan, nurse)*<br>*"As far as I remember, sugar was not checked frequently, however, blood pressure was continuously monitored" (Pakistan, Carer)*<br>*"BP targets were difficult in some, depending on their (patients) response to labetalol but were met as early as possible. INR was difficult due to unavailability of reports as early as we would like it to be ready" (Sri Lanka, Neurologist)*<br>*"PCR result has not yet been confirmed, subject stayed in an isolated area with limited access. It was hard to record the blood pressure and temperature as protocol required frequency" (Vietnam, Neurologist)* |

BGL denotes blood glucose level, BP blood pressure, ICH intracerebral haemorrhage

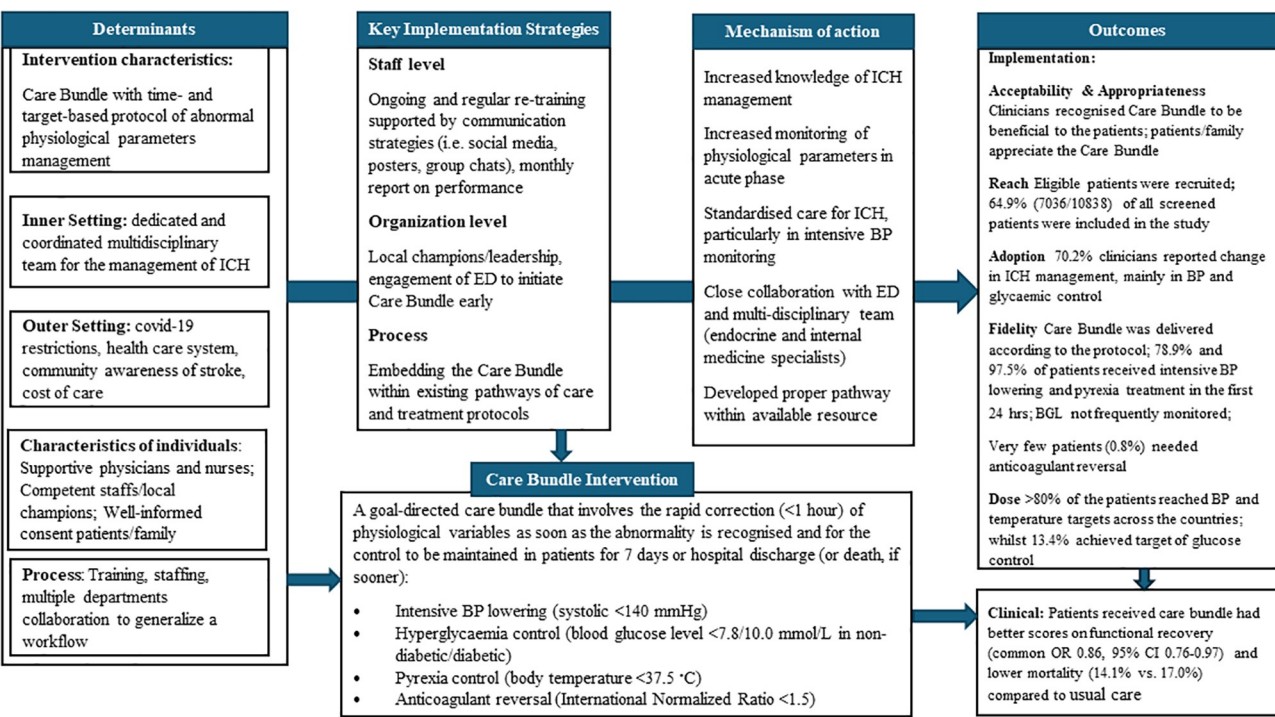

**Fig 1. Synthesis findings by implementation research logic model.** BGL blood glucose level, BP denotes blood pressure, CI confidence interval, ED Emergency Department, ICH intracerebral haemorrhage, OR odds ratio.

intervention due to increased workload from the need for intensive monitoring. Conversely, patients and carers felt reassured by the frequent monitoring by healthcare providers (Table 1).

## Reach, adoption, fidelity & dose

Among 10838 screened ICH patients, 7036 (64.9%) were included in the INTERACT3 study. Reasons for non-inclusion were recorded in screening logs, with the predominant reason (> 50% of excluded patients) being delayed presentation of >6 hours since symptom onset across all countries (S3 Table). From the interviews, the reach of eligible patients was influenced by prehospital factors such as stroke awareness and access to emergency care, shortage of beds and staff, and difficulty in obtaining consent during an acute event issue. All implementers reported that all physiological parameters were readily monitored in the participating patients (Table 1). Intensive BP lowering and anti-pyrexia treatment were delivered well within the first 24 hours (S4 Table and S2 Fig), with >80% of the patients reaching the SBP target in most of the countries, whilst achieving the glucose control target was invariably low (from 0%-40% across countries; S3 Fig). From our survey, about a third of respondents expressed difficulty in achieving the targets for SBP, glucose and anticoagulant reversal (S4 Fig). Challenges in timely initiation of the Care Bundle included the additional time required for PCR testing and waiting for its results during the COVID-19 pandemic, and limited availability of intravenous antihypertensive medication and the nature of critical conditions of patients to reach the targets. The main barriers to achieving the SBP target were shortage of medication and concerns over adverse effects (S5 Fig). While concerns over hypoglycemia,

difficulty in controlling blood glucose, and reluctance to start insulin infusion for intense glycemia control are the main barriers to achieving the glycemic target (S6 Fig).

## Mechanisms of integration

From the quality control survey of 47 respondents, 70.2% of respondents reported that their treatment and management changed after participating in the trial, particularly the focus on early intensive BP monitoring (S5 Table). Key mechanisms according to NPT domains of integrating the Care Bundle into routine practice are summarised in Table 2 with corresponding quotes illustrated in S6 Table.

Under the domain of coherence, clinicians perceived the potential benefits of the Care Bundle in improving patient outcomes, though their commitment was impacted by the requirement to adhere to the strict protocol in their clinical management and the increased workload. Under the domain of collective participation, leveraging additional equipment and medications through participation in the trial, was a strong motivative factor in the context of limited resources. Under the domain of collective action, understanding how to embed the Care Bundle within existing pathways of care and treatment protocols was crucial. Ongoing and regular re-training supported by communication strategies (social media, posters) was necessary to maintain the implementation. Successful implementation of the Care Bundle required close collaboration across the Emergency Department (ED) and a multi-disciplinary team with input from endocrine and internal medicine specialists. Under the domain of reflexive monitoring, participants described the need to increase community awareness of stroke symptoms and to address local health system context such reducing patient out-of-pocket costs, sufficient

**Table 2. Themes for integrating the Care Bundle into routine care by normalisation process theory (NPT) domains.**

| NPT domains and coded themes | Explanation |
| --- | --- |
| *Coherence* | • Good understanding of the study rationale<br>• Appreciate the Care Bundle and perceive a benefit in participating in the trial<br>• Usual care is easier/simpler since the Care Bundle is intensive and strict |
| *Cognitive participation* | • In the context of limited resources, the conduct of the trial made it possible for the hospital to receive more equipment and medications for patient care<br>• Clinicians perceived the benefits of Care Bundle to ICH patients<br>• Increased workload for intensive monitoring influences the commitment |
| *Collective action* | • It was important to develop a proper pathway, adapt treatment protocols and have sufficient learning materials to support the implementation of the intervention into practice<br>• Multiple departments are required to collaborate and there is a need for multidisciplinary care to support the implementation of Care Bundle |
| *Reflexive monitoring* | • Constant and regular basis training helps maintain the Care Bundle implementation, particularly in the ED<br>• Detailed guidance and policy are important for integration and sustainability<br>• Increase prehospital awareness, patient, and community symptom identification<br>• Recognised the importance of adopting into routine care and widely used, but factors such as resources, knowledge and admit of the benefits of the Care Bundle need to be solved to ensure the sustainability |

BGL denotes blood glucose level, BP blood pressure, ED emergency department, ICH intracerebral haemorrhage, PI principal investigator

equipment (e.g. infusion pumps, imaging) and training workforce (e.g. standard stroke proto-cols/pathways in ED).

## Sustainability and contextual barriers

As outlined in Table 3 with illustrative quotes, participants reported several contextual barriers that influenced the sustainable implementation of the Care Bundle that varied across countries, including the economic burden for care, shortage of resources, and barriers to initiating timely care during the COVID pandemic. In Latin America, the health system and infrastructure-related challenges impacted the fidelity and dose of the care bundle implementation: there was a lower proportion of patients reached SBP<140 mmHg compared to the overall average (92% vs. 97%) (S3 Fig vs. S4 Table) due to the high cost and low availability of the anti-hypertensive agents; and shortage of the nursing staff resources to monitoring the BGL resulted only 13% of patients reaching the BGL targets (S3 Fig). The sociocultural factor of patients' health seeking behaviour influenced the reach and recruitment of the eligible participants in India and Nigeria. For example, in Nigeria, all the excluded patients were due to delayed admission to the hospital (S3 Table) which related to the poor community awareness of stroke symptoms, and beliefs in witchcraft or herbal therapy meant that patients/their families preferred these avenues for treatment rather than seeking immediate attention at a hospital. In Vietnam, the lack of medications to deliver the care bundle had impacted the implementation outcome of fidelity. In Pakistan, health system related factors such as high out-of-pocket costs of treatment and shortage of staffing resources had impacted the fidelity of the implementation, showing a lower proportion of patients received treatment for glucose control (20%) and pyrexia (56%) compared to the other countries (S2 Fig). The insurance and the financial burden of the treatment also influenced the uptake of the treatment by patients: there were 27% of the participants refused to give consent to participate the study (S3 Table). In Latin America, Sri Lanka and Vietnam, healthcare infrastructure factors such as delays in the process of diagnosis and limited resourcing (e.g. staff, space) during the COVID pandemic were the main barriers to recruiting eligible patients and delivering the care bundle, which impacted upon the implementation outcomes of reach and fidelity. For example, there were a lower percentage of participants in Chile/Peru, and for those excluded patients in Peru were mainly due to delayed admission (>6 hrs from stroke onset) compared to overall average (85.2% vs. 67.5%). A summary of the contextual barriers that impeded the care bundle implementation by country is reported in Table 3.

## Discussion

The embedded PE in INTERACT3 has shown that the Care Bundle was generally delivered as planned and accepted by the stakeholders. In comparison to our findings of the PE in China [14], this study has highlighted the significant adverse impact of the COVID-19 pandemic in other countries on the Care Bundle implementation due to slow recruitment, prolonged consent process and shortage of staffing. Findings in China indicate that BP lowering is already part of routine ICH management but with a wide range of SBP targets applied in the first 24 hours. In contrast, we found there were no standard protocols for BP management for routine care in other countries compared to China. This explains why the differences in BP control between the Care Bundle and usual care groups were larger and the effect of the Care Bundle was greater, in countries outside of China [9]. Though concerns about the potential harms of intensive BP lowering and challenges to reaching the physiological targets were common in China and other LMICs, the interviewees from outside of China emphasized that sufficient and constant training was essential to support the persistent adoption of the Care Bundle into

**Table 3. Contextual factors according to participating countries.**

| Country | CFIR Domain and construct | Contextual barriers | Illustrative quotes | Implementation outcomes impacted upon |
|---|---|---|---|---|
| Chile/ Peru/ Mexico | Outer setting: local conditions- Healthcare system related financial | High cost of the medication | *"The intravenous antihypertensive agent alone, is very high cost, so it's not available for all the patients." (Focus group, Chile)* | Fidelity |
| | Inner setting: available resources | Limited accessibility of the medication | *"We have limited access to intravenous drugs for blood pressure control. . .made it difficult for us to reach the blood pressure goal."(Mexico, Neurologist)* | Dose |
| | Individual: Recipients-Sociocultural | Pandemic resulted in late presentation due to patients/families fear of going to hospitals | *"When the pandemic, the fear appeared in patients. . . and the hospital policies influence limited the recruitment, and the resource is limited." (Peru, Focus group)* | Reach |
| | Inner setting: healthcare infrastructure | Shortage of workforce | *"We did have some problems with the nursing staff that sometimes—related to glycaemic control." (Mexico, Neurologist)* | Dose |
| Nigeria | Inner setting: available resources | Limited resources for diagnosis | *"I think some hospitals might not have the neuro-imaging techniques necessary to make the diagnosis." (Nigeria, Neurologist)* | Reach |
| | Inner setting: available resources | Short of staff resourcing due to hospital strike during COVID pandemic | *"We had a major disruption and it prevented us from recruiting patients for more than two months because our resident doctors were on strike. During that period, we couldn't get patients to recruit." (Nigeria, focus group)* | Reach |
| | Individual: Recipients-Sociocultural | Late admission due to lack of community awareness of stroke and culture/beliefs of patients about seeking treatment of stroke | *"Most of our patients will go to the herbalist, our community because they do not believe it is God's wish, they will believe it is witch or wizard, so they won't come to the hospital immediately." (Nigeria, nurse)*<br>*"At the patient level, we have significant delays in getting to the hospital on the part of the patients. . ..." (Nigeria, focus group)* | Reach |
| Vietnam | Inner setting: work infrastructure | Pandemic caused workforce shortage and a delayed process in diagnosis | *"The pandemic is the biggest barrier affected the recruitment and the implementation. . .there is a shortage of human resources in the department" (Vietnam, Neurologist)*<br>*"Subject was admitted but stayed in ED for about 6–8 hours until the PCR test confirmed. . .It's hard to ask for operation from ED staff since their workloads were high." (Vietnam, Neurologist)* | Reach |
| | Inner setting: available resources | Medication limitation for care bundle delivery | *"With the reversal of anticoagulation, there is a big difficult when FFP and PPC are not always available in the hospital" (Vietnam, Neurologist)*<br>*"Hospitals that do not have nicardipine or equipment or drugs to manage blood glucose. . .room conditions might not be adequate." (Vietnam, Neurologist)* | Fidelity |
| Pakistan | Outer setting: local conditions- Health care system related financial | Health care system related financial issues due to insurance cover | *"There is also an issue of affordability between private and government hospitals. . .their monitoring is sometimes not possible because they have to go to another hospital for their complete management." (Pakistan, Neurologist)*<br>*"Family's financial is a big issue, some have affordability issue to use medication." (Pakistan, Neurologist)* | Reach and Fidelity |
| | Inner setting: available resources | Work overload | *"Other issues are staff resources related, as discussed earlier they are busy, and they are overloaded as a result" (Pakistan, Nurse)* | Fidelity |
| Sri Lanka | Inner setting: available resources | Unavailable to report INR due to lack of lab resources | *INR was difficult due to unavailability of reports as early as we would like it to be ready" (Sri Lanka, Neurologist)* | Dose |
| | Individual: Recipients—sociocultural | Pandemic caused workforce shortage | *"Due to the COVID pandemic, our ward staff and space facility is limited, such as the availability of high dependency units and monitoring." (Sri Lanka, Neurologist)* | Reach |

*(Continued)*

**Table 3.** (Continued)

| Country | CFIR Domain and construct | Contextual barriers | Illustrative quotes | Implementation outcomes impacted upon |
|---|---|---|---|---|
| India | Individual: Recipients—sociocultural | Delayed admission to hospitals due to patients' health seeking behaviour to seek treatment from local doctors | *"We should increase awareness of stroke in the community first . . . to receive the stroke patients without delay." (India_002)*<br>*"Recruiting patients was difficult because most of the patients go to local doctors and there's a lot of delay in the tertiary care hospital" (India, Neurologist)* | Reach |

CFIR denotes Consolidated Framework for Implementation Research, ED emergency department, FFP fresh frozen plasma, PPC prothrombin complex concentrate

routine practice. The suboptimal control of glycemia with low rate of achieving the target was noted across all the countries. Most neurologists are not trained and comfortable using insulin infusion, consultation and collaboration with endocrinology to improve the insulin treatment will enhance the implementation of the care bundle.

Medication limitation was a common barrier to deliver the care bundle, mainly due to high cost of the medication and insufficient supply at the hospital. Despite contextual factors related to staffing and medication shortage that were identified previously in China [14], additional health system barriers included financial burden from the costs of care, delayed hospital arrival from poor public awareness of stroke, and prolonged length of stay in the emergency department were reported in other LMICs. Although access to essential medications is part of the right to health care as noted by the Sustainable Development Goals [22]; and at least one type of antihypertensive agents, insulin, and oral hypoglycemic drugs are reported available, affordable, or endorsed in Africa [23], medications resourcing and cost issues were noted in our study. Strategies to promote the uptake of the Care Bundle to improve ICH management in LMICs are not limited to educating clinicians and other healthcare providers about protocols and reducing the clinical concerns of the new care model, but must also involve diverse disciplines, from health policy and pharmaceutical to economics to address access to care [24].

Besides the sociocultural beliefs that impede seeking treatment in acute medical care facilities in Nigeria, the scarcity of culturally appropriate materials about stroke symptoms, and underutilization of ambulance services for stroke patients are known barriers to effective access to stroke services in LMIC [25, 26]. Therefore, further ensuring the Care Bundle is maximally effective requires education of the public about stroke symptoms and timely admission to the hospital to increase the utilization of stroke services in LMICs. Establishing standard stroke protocols that are tailored to regional and local needs can reduce time lost in transit/ED and enhance the delivery of stroke care [27]. Herein, we advocate the widespread adoption of the Care Bundle into the local standard protocol for stroke care, with coordinated interdisciplinary care to optimise the chances of patients globally having improved functional recovery after suffering an ICH.

## Next steps

Scaling up the implementation of the INTERACT3 Care Bundle is required to optimize outcomes for patients with acute ICH globally. We believe a range of implementation strategies and dissemination activities are required to promote the uptake of the Care Bundle around the world, with an iterative process with the sharing of lessons learnt. The ability to rapidly transform services to consolidate the Care Bundle for active ICH management and improve quality

of care needs facilitation by collaborations and partnerships across the global stroke community and organizations, which aligns with the World Stroke Organisation Roadmap and guidelines for delivering quality stroke care [28]. The dissemination activities should also include campaigns to increase public awareness of stroke and the need to rapidly seek hospital treatment. Before scaling up the Care Bundle in real-world settings, an assessment of the local health system context such as health financing, medication, workforce, and infrastructure is essential to develop locally appropriate pathways to facilitate the implementation.

## Strength and limitations

Scale-up of interventions is challenging due to the complexity of implementation across diverse contexts with differing populations, finances, resources, and capacity [29, 30]. This PE alongside the INTERACT3 is unique done in diverse health care systems across multiple countries, where the findings will promote the implementation of care bundles in day-to-day practice. It also allows a better understanding of how to optimize implementation of the Care Bundle across LMIC settings and will provide background and context information for future research. Although we were able to undertake interviews with patients/carers and nurses as planned, the number of participants available for interviews was limited, and not balanced by country due to constraints imposed by the COVID-19 pandemic. Although we made efforts to include all the participating sites to complete the survey, we acknowledge the limitation that some of the countries had no or very few sites to participate the survey, which might cause volunteer bias. Due to a limited number of secondary hospitals involved in INTERACT3, the study was undertaken at academic tertiary-referral level hospitals with more capacity and greater interest in research, and may not be transferable to other settings.

## Conclusions

The INTERACT3 PE has provided insights into the challenges of implementation of the goal-directed Care Bundle in different contexts. The multiple barriers include staff shortage, limited availability of antihypertensive drugs, and delayed systems of care processes that influence implementation need to be solved to enhance the wide application of the Care Bundle in clinical practice.

## Supporting information

**S1 Table. Percentage of participating sites for survey and interviews by country.**
(DOCX)

**S2 Table. Characteristics of the purposive sampled interview participants.**
(DOCX)

**S3 Table. Reason of exclusion by country in screened participants.**
(DOCX)

**S4 Table. Reach and dose of patients for the care bundle implementation.**
(DOCX)

**S5 Table. Treatment changes for ICH management by country from the survey results.**
(DOCX)

**S6 Table. Embedding the Care Bundle into routine care by normalisation process theory (NPT) domains.**
(DOCX)

**S1 Fig. Characteristics of participants in quality control survey.**
(TIF)

**S2 Fig. Care Bundle delivery during the first 24 hours.**
(TIF)

**S3 Fig. Target reached for BP lowering and BGL control.**
(PNG)

**S4 Fig. Implementers' perceptions of achieving the intervention targets from the survey.**
(TIF)

**S5 Fig. Joint analysis of survey and interview on intensive BP lowering barriers.**
(JPG)

**S6 Fig. Joint analysis of survey and interview on glycemia control barriers.**
(JPG)

**S1 Checklist. Standards for reporting qualitative research (SRQR) checklist.**
(DOCX)

## Author Contributions

**Conceptualization:** Menglu Ouyang, Lili Song, Hueiming Liu.

**Data curation:** Anila Anjum, Francisca Gonzalez Mc Cawley.

**Formal analysis:** Menglu Ouyang, Anila Anjum, Francisca Gonzalez Mc Cawley.

**Investigation:** Mohammad Wasay, Paula Muñoz Venturelli, H. Asita de Silva, Nguyen Huy Thang, Kolawole W. Wahab, Jeyaraj D. Pandian, Octavio M. Pontes-Neto, Carlos Abanto, Venessa Cano-Nigenda, Antonio Arauz, Chao You.

**Methodology:** Menglu Ouyang, Hueiming Liu.

**Project administration:** Xiaoying Chen, Alejandra Malavera, Paula Muñoz Venturelli, H. Asita de Silva, Nguyen Huy Thang.

**Resources:** Lu Ma, Xin Hu, Xi Li, Chao You.

**Supervision:** Stephen Jan, Craig S. Anderson, Hueiming Liu.

**Visualization:** Menglu Ouyang.

**Writing – original draft:** Menglu Ouyang, Hueiming Liu.

**Writing – review & editing:** Menglu Ouyang, Mohammad Wasay, Lu Ma, Xin Hu, Xiaoying Chen, Alejandra Malavera, Xi Li, Paula Muñoz Venturelli, H. Asita de Silva, Nguyen Huy Thang, Kolawole W. Wahab, Jeyaraj D. Pandian, Octavio M. Pontes-Neto, Carlos Abanto, Venessa Cano-Nigenda, Antonio Arauz, Chao You, Stephen Jan, Lili Song, Craig S. Anderson, Hueiming Liu.

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
