## [Decision Letter · Decision Letter 0]

9 Aug 2024

PGPH-D-24-01251

Implementation of a goal-directed care bundle for intracerebral hemorrhage: results of embedded process evaluation in the INTERACT3 trial

Dear Dr. Ouyang,

Thank you for submitting your manuscript to PLOS Global Public Health. After careful consideration, we feel that it has merit but does not fully meet PLOS Global Public Health’s publication criteria as it currently stands. Therefore, we invite you to submit a revised version of the manuscript that addresses the points raised during the review process.

We look forward to receiving your revised manuscript.

Kind regards,

Buna Bhandari

Academic Editor

Journal Requirements:

2. Please provide separate figure files in .tif or .eps format.

3. PLOS GPH considers qualitative and mixed-methods studies for publication. We recommend that authors use the COREQ checklist, or other relevant checklists listed by the Equator Network, such as the SRQR, to ensure complete reporting (http://journals.plos.org/globalpublichealth/s/submission-guidelines#loc-qualitative-research). In general, we would expect qualitative studies to include the following: 1) defined objectives or research questions; 2) description of the sampling strategy, including rationale for the recruitment method, participant inclusion/exclusion criteria and the number of participants recruited; 3) detailed reporting of the data collection procedures; 4) data analysis procedures described in sufficient detail to enable replication; 5) a discussion of potential sources of bias; and 6) a discussion of limitations. 

4. Authors conducting research in other countries or with Indigenous populations are required to complete a copy of PLOS’ questionnaire on inclusivity in global research. The policy applies to researchers who have travelled to a different country to conduct research, research with Indigenous populations or their lands, and research on cultural artefacts. You can find more information on this policy here: https://journals.plos.org/globalpublichealth/s/best-practices-in-research-reporting

Additional Editor Comments:

Thank you for submitting your interesting manuscript. Please address the following concerns in your revision:

In the methods section, three focus group discussions are mentioned as a data collection method, but details about the FGDs are missing, such as how many participants were in each FGD and where (which country) they were conducted. Please provide these details.

Secondly, although a separate publication from the same study site in China is mentioned in the paper, it is still unclear why the findings from all the countries were not presented together. How do the methods and findings of this paper differ from the study conducted in China, given that it is the same trial site? Additionally, how is a separate publication justified without issues of data slicing? Please clarify your approach clearly.

Reviewers' comments:

Reviewer's Responses to Questions

**Comments to the Author**

1. Does this manuscript meet PLOS Global Public Health’s publication criteria? Is the manuscript technically sound, and do the data support the conclusions? The manuscript must describe methodologically and ethically rigorous research with conclusions that are appropriately drawn based on the data presented.

Reviewer #1: Yes

Reviewer #2: Yes

2. Has the statistical analysis been performed appropriately and rigorously?

Reviewer #1: Yes

Reviewer #2: N/A

3. Have the authors made all data underlying the findings in their manuscript fully available (please refer to the Data Availability Statement at the start of the manuscript PDF file)?

Reviewer #1: Yes

Reviewer #2: Yes

4. Is the manuscript presented in an intelligible fashion and written in standard English?

Reviewer #1: Yes

Reviewer #2: Yes

5. Review Comments to the Author

Reviewer #1: The manuscript meets all the requirements of the Journal, is well-written, interesting and contains the necessary elements for the reproduction of the research undertaken by the authors. An additional advantage of the work is its applied purpose.

I recommend publication in its present form.

Reviewer #2: Thank you for the opportunity to review this manuscript reporting on a process evaluation embedded in a stepped-wedge, cluster-randomized, Intensive Care Bundle with Blood Pressure Reduction in Acute Cerebral Hemorrhage Trial (INTERACT3). The reported process evaluation was conducted in 7 LMIC. A total of 27 interviews (16 doctors, 3 nurses, and 8 patients/carers) and 3 focus group discussions were conducted, alongside 47 responses from the quality control survey at 25 hospitals. The manuscript is well written and presents relevant findings. However, I believe that there are a number of points that should be addressed by the authors.

1. The introduction is very light on details regarding ICH and background information on why this multifaceted care bundle was selected. I would appreciate some more literature and context.

2. The major limitation of the paper is the focus on a range of countries (7 to 10 - this is somewhat unclear, see my next comment) for the process evaluation. The authors previously published the process evaluation conducted in China - why were all of the other countries grouped together? What is the rationale? What are the implications of combining all of the data across countries in a single paper? This raises other issues discussed below.

3. How many countries were included in the process evaluation? At one point, it is mentioned that "A quality control survey was collected to understand the experience and project operational barriers to implementing the intervention at sites in Brazil, Chile, India, Mexico, Nigeria, Pakistan, Peru, Sri Lanka, and Vietnam" whereas later it is mentioned "11 sites agreed to participate from India, Sri Lanka, Vietnam, Pakistan, Chile, Nigeria and Mexico." How many countries were included in this process evaluation, and how many sites across these countries (how many per country?)? This is also complicated by the fact that data is present from countries that should be absent from this paper (e.g., China data in Table S2 - Supposed to be in its own paper and already published).

4. How was the interview guide elaborated? Was it based on a theoretical model? What were the questions? What the interview guide pilot tested? The interview guide should be attached as an additional file.

5. While Table 1 is interesting, there are a number of issues. First, some implementation outcomes are grouped together (acceptability and appropriateness). While I understand that these can be conceived as being similar, I believe there would be value in separating these out like the other outcomes. Second, and this is fundamental, there is a need to present the results for each implementation outcome for each country. There is little value in present an overall general theme and some quotes for some of the countries, given that all countries would have widely different contexts. Consequently, the authors should re-work and expand Table 1 to present data for each implementation outcome for each country.

6. Table 2 is a similar situation; it should present country-specific information. I think it is fine to have a general summary/explanation for each NPT domain, but without the granularity of each country the paper loses some of its appeal.

7. Table 3 could be retrustructed to align with the domains and/or the contracts of the Consolidated Framework for Implementation Research (CFIR): Damschroder, L.J., Reardon, C.M., Widerquist, M.A.O. et al. The updated Consolidated Framework for Implementation Research based on user feedback. Implementation Sci 17, 75 (2022). https://doi.org/10.1186/s13012-022-01245-0

8. The discussed is written nicely. It would be relevant to discuss in more detail similarities and discrepancies across countries in terms of implementation outcomes, mechanisms of integration and contextual factors.

6. PLOS authors have the option to publish the peer review history of their article (what does this mean?). If published, this will include your full peer review and any attached files.

**Do you want your identity to be public for this peer review?** For information about this choice, including consent withdrawal, please see our Privacy Policy.

Reviewer #1: No

Reviewer #2: No

---

## [Decision Letter · Decision Letter 1]

2 Oct 2024

PGPH-D-24-01251R1

Implementation of a goal-directed care bundle for intracerebral hemorrhage: results of embedded process evaluation in the INTERACT3 trial

Dear Dr. Ouyang,

Thank you for submitting your manuscript to PLOS Global Public Health. After careful consideration, we feel that it has merit but does not fully meet PLOS Global Public Health’s publication criteria as it currently stands. Therefore, we invite you to submit a revised version of the manuscript that addresses the points raised during the review process.

Thank you for addressing most of the comments in the revised manuscript. However, following questions raised previously has not been addressed. Please address the following concerns along with the reviewer's comments in your revision:

In the methods section, three focus group discussions are mentioned as a data collection method, but details about the FGDs are missing, such as how many participants were in each FGD and where (which country) they were conducted. Please provide these details.

Secondly, although a separate publication from the same study site in China is mentioned in the paper, it is still unclear why the findings from all the countries were not presented together. How do the methods and findings of this paper differ from the study conducted in China, given that it is the same trial site? Additionally, how is a separate publication justified without issues of data slicing? Please clarify your approach clearly.

We look forward to receiving your revised manuscript.

Kind regards,

Dr Buna Bhandari

Academic Editor

Additional Editor Comments (if provided):

Thank you for addressing most of the comments in the revised manuscript. However, following questions raised previously has not been addressed. Please address the following concerns along with the reviewer's comments in your revision:

In the methods section, three focus group discussions are mentioned as a data collection method, but details about the FGDs are missing, such as how many participants were in each FGD and where (which country) they were conducted. Please provide these details.

Secondly, although a separate publication from the same study site in China is mentioned in the paper, it is still unclear why the findings from all the countries were not presented together. How do the methods and findings of this paper differ from the study conducted in China, given that it is the same trial site? Additionally, how is a separate publication justified without issues of data slicing? Please clarify your approach clearly.

Reviewers' comments:

Reviewer's Responses to Questions

**Comments to the Author**

1. If the authors have adequately addressed your comments raised in a previous round of review and you feel that this manuscript is now acceptable for publication, you may indicate that here to bypass the “Comments to the Author” section, enter your conflict of interest statement in the “Confidential to Editor” section, and submit your "Accept" recommendation.

Reviewer #2: (No Response)

2. Does this manuscript meet PLOS Global Public Health’s publication criteria? Is the manuscript technically sound, and do the data support the conclusions? The manuscript must describe methodologically and ethically rigorous research with conclusions that are appropriately drawn based on the data presented.

Reviewer #2: Yes

3. Has the statistical analysis been performed appropriately and rigorously?

Reviewer #2: N/A

4. Have the authors made all data underlying the findings in their manuscript fully available (please refer to the Data Availability Statement at the start of the manuscript PDF file)?

Reviewer #2: No

5. Is the manuscript presented in an intelligible fashion and written in standard English?

Reviewer #2: Yes

6. Review Comments to the Author

Reviewer #2: Thank you to the authors for revising the manuscript. I appreciate the enhancements made to the manuscript to address my comments. However, I still have a number of points that I believe should be addressed.

1. Although the authors have added more information to the introduction, it still does not fully establish the context and rationale for the specific interventions used in the care bundle. There should be a clearer link between the choice of components in the bundle and their specific relevance to the pathophysiology of ICH in LMICs. For example, more details on the existing standard of care in the included countries and how the care bundle adds value would help strengthen the justification for the study.

2. While I understand the logistical and design challenges of conducting a multi-country evaluation, simply grouping the findings from all non-China LMIC sites together may obscure critical contextual differences. For a process evaluation, understanding the local implementation context is crucial. I suggest that the authors provide a more nuanced comparison, highlighting specific variations in implementation across the countries (e.g., differences in healthcare infrastructure, sociocultural factors, or policy environments).

3. The use of the term “vanguard phase” to describe the separate process evaluation for China is still a bit ambiguous. It would be helpful if the authors clearly articulate the purpose of this phase, its specific objectives, and why the decision was made to analyze these data separately. Additionally, they should discuss how findings from the China phase were used to inform implementation in the subsequent countries.

4. The authors have added some country-specific information in supplementary tables, but these are not integrated into the main text, making it difficult to interpret the contextual nuances of implementation across countries. A more thorough country-specific comparison within the main text would add significant value to the paper and support the argument that implementation is context-dependent.

5. The restructuring of Table 3 based on the CFIR framework is a step in the right direction. However, the information remains too general. Each CFIR domain should have clearer links to how specific contextual factors influenced implementation success or failure in each country. Additionally, illustrative quotes should be tied more directly to the relevant constructs to make the table more informative.

I believe addressing these points would add value to the manuscript and render it suitable for publication.

7. PLOS authors have the option to publish the peer review history of their article (what does this mean?). If published, this will include your full peer review and any attached files.

**Do you want your identity to be public for this peer review?** For information about this choice, including consent withdrawal, please see our Privacy Policy.

Reviewer #2: No

---

## [Editor Report · Decision Letter 2]

22 Nov 2024

Implementation of a goal-directed care bundle for intracerebral hemorrhage: results of embedded process evaluation in the INTERACT3 trial

PGPH-D-24-01251R2

Dear Dr Ouyang,

We are pleased to inform you that your manuscript 'Implementation of a goal-directed care bundle for intracerebral hemorrhage: results of embedded process evaluation in the INTERACT3 trial' has been provisionally accepted for publication in PLOS Global Public Health.

Best regards,

Dr Buna Bhandari

Academic Editor